# The Relationship between Serum Uric Acid and Ejection Fraction of the Left Ventricle

**DOI:** 10.3390/jcm10174026

**Published:** 2021-09-06

**Authors:** Ivan Vlad-Sabin, Buzas Roxana, Cuțina Morgovan Adina-Flavia, Ciubotaru Paul, Ardelean Melania, Goje Daniel, Roșca Ciprian Ilie, Timar Romulus, Lighezan Daniel

**Affiliations:** 1Advanced Research Center for Cardiovascular Pathology and Haemostaseology, Department of Internal Medicine I-Medical Semiology I, “Victor Babes” University of Medicine and Pharmacy, 300041 Timisoara, Romania; vlad_sabin_ivan@yahoo.com (I.V.-S.); adina.morgovan@yahoo.com (C.M.A.-F.); paulciubotaru@gmail.com (C.P.); ardelean.melania@gmail.com (A.M.); dan.i.goje@gmail.com (G.D.); roscaci@yahoo.com (R.C.I.); dlighezan@gmail.com (L.D.); 2Nutrition and Metabolic Diseases, Department of Internal Medicine II-Diabetes, “Victor Babes” University of Medicine and Pharmacy, 300041 Timisoara, Romania; timarrz@yahoo.com

**Keywords:** serum uric acid, hyperuricemia, heart failure, ejection fraction

## Abstract

**Study basis:** As a byproduct of protein metabolism, serum uric acid is a controversial risk factor and is the focus of several recent studies in the field of cardiovascular disease. Whether serum uric acid is involved in the development of these pathologies alone or in conjunction with other factors is a matter of debate. **Objective:** The objective of this study is to assess the direct relationship between serum uric acid and the ejection fraction. **Methods:** A retrospective study of 303 patients with heart failure, classified according to the ESC guidelines, was conducted, and several parameters, along with the relationship between serum uric acid and ejection fraction, were characterized. **Results:** A direct relationship between the level of serum uric acid and the ejection fraction was established (*p =* 0.03); patients with higher uric acid had an increased risk of having a lower ejection fraction. **Conclusions**: Serum uric acid, even when asymptomatic, is linked with the level of the ejection fraction of the left ventricle.

## 1. Introduction

Heart failure (HF), irrespective of the clinical phenotype, is a leading cause of morbidity and mortality worldwide. It increases with age and is aggravated by numerous risk factors. In the latest AHA report on cardiovascular diseases published in 2021 [1], HF is estimated to increase by 46% by 2030, thus affecting almost 3% of the adult population.

There is evidence that the prevalence of reduced ejection fraction heart failure (refHF) is decreasing, with a stabilization of mid-range ejection fraction heart failure (mrHF) prevalence and an increasing number of patients with preserved ejection heart failure (pefHF) [2]. The trend of decreasing incidence of refHF is due to multiple factors, including the improvement of cardiovascular therapies, better access to healthcare, improved revascularization techniques, a reduced time from event to stent placement and public campaigns to raise awareness. Factors contributing to the increasing number of patients with symptoms consistent with pefHF are an aging population, increasing obesity rates, a steady increase in metabolic syndrome and diabetes mellitus cases, a western style diet high in carbohydrates and protein, and low physical activity levels.

One of the risk factors for cardiovascular diseases is uric acid levels and hyperuricemia [3], with established data for prognosis and increasing evidence that uric acid is an independent risk factor for the development of cardiovascular diseases, even where the patient does not have levels of serum uric acid (SUA) consistent with gout [4]. Uric acid is also mentioned as a routine parameter that needs to be checked when assessing the cardiovascular risk of a patient [5].

## 2. Objective

The purpose of this study is to evaluate the relationship between SUA and HF, as classified according to the European Society of Cardiology guidelines, in a region of Europe that is characterized by a high prevalence of cardiovascular disease, obesity, and diabetes mellitus.

## 3. Materials and Method

We retrospectively analyzed a series of 303 consecutive patients with known chronic heart failure, over a period of 6 months, admitted to an Emergency Internal Medicine Department in a city in western Romania. All patients presented themselves or were transported by ambulance to the ER, where they were evaluated and admitted to the Internal Medicine Department with decompensated chronic heart failure. Patients were divided into three groups based on their ejection fraction: the first group included patients with pefHF, the second group included patients with mrHF, and the third group included patients with refHF.

During their stay, all patients underwent cardiac ultrasound, performed by an experienced cardiologist. Planimetric ejection fractions and left atrial volume were measured using a General Electric GE Vivid E9 Ultrasound System.

A history of diabetes mellitus, arrhythmias, and chronic coronary syndromes were taken into consideration, including for patients who died during admission, along with the chronic cardiac medication administered according to the ESC guidelines.

Blood samples were taken, and measurements of several biochemical parameters were performed, including serum electrolytes (Na^+^ and K^+^), serum creatinine (SC), and a lipid panel that included LDL cholesterol (LDLc), HDL cholesterol (HDLc), total cholesterol (TC), triglycerides (TG), and serum uric acid (SUA). The estimated glomerular filtration rate was calculated using an MDRD creatinine equation. Patients with an estimated glomerular filtration rate of <15 mL/min/1.73 m^2^ were included in our analysis. SUA was measured from plasma using a Dimension Integrated Chemistry System and reported in mg/dL. Patients with already known hyperuricemia, gout, and chronic use of serum uric acid lowering agents or thiazide diuretics, as well as patients on chronic dialysis, were excluded from the study. Other exclusion criteria were solid organ neoplasms, malignant or chronic hematological diseases, autoimmune conditions, and immunosuppressive and biological therapies. In addition, patients that were intubated or were resuscitated after sudden cardiac arrest in the ER were not included in our analysis.

NT-proBNP levels were also measured for all patients during the first 24 h after admission, from blood samples, using a MINI VIDAS^®^ compact multiparametric immune-analyzer, with results expressed in pg/mL.

Arrhythmias were also taken into consideration when analyzing our data. Only supraventricular arrythmias were documented, and the following were included in our analysis: atrial fibrillation, atrial flutter, and supraventricular extrasystoles. Patients with malignant arrhythmias such as ventricular tachycardia or other ventricular electric disturbances were excluded. Patients with implanted defibrillators or pacemakers were also excluded from our study.

## 4. Statistical Analysis

Statistical analysis was based on ANOVA one-way tests for continuous variables and Chi-square tests for categorical variables, with the corresponding *p*-values being presented in the summary tables. Pairwise comparisons were performed based on Bonferroni-adjusted significance tests, with the corresponding significance (S; *p* ≤ 0.05) or non-significance (NS; *p >* 0.05) being reported. *p*-values based on Bonferroni correction comparison for comparing the FEVS >50% category with the FEVS 49–40% category patients are flagged under *p*-value 1 column in the summary tables, while *p*-value 2 flags the comparison results of the FEVS >50% category with the FEVS <40% category patients and *p*-value 3 flags the comparison results of the FEVS 49–40% category with the FEVS <40% category patients.

Single linear regression analysis was applied to assess the link between NT-proBNP, uric acid, and LVEF.

## 5. Results

Patient baseline characteristics are summarized in Table 1, comorbidities are summarized in Table 2, and main laboratory biochemical parameters are summarized in Table 3.

We obtained significant statistical results when we compared the baseline characteristics of the patients analyzed. A significant statistical difference was found in the mean value of age between patients in the FEVS >50% and FEVS <40% categories. Patients in the FEVS >50% category were older, as compared to the FEVS <40% category patients. In addition, a significant statistical difference (*p* < 0.001) in the proportion of males/females among the three groups of patients was observed.

A significant difference among the proportion of patients having NYHA class II in the FEVS >50% vs. the FEVS <40% group of patients (i.e., significantly higher proportion of patients with NYHA class II in the FEVS >50% category as compared to the FEVS <40% category) was also observed.

A significant statistical difference (*p* = 0.003) in the proportion of patients with a history of diabetes among our three groups of patients was observed, with a significant difference in the FEVS >50% vs. the FEVS <40% group of patients (i.e., significantly higher proportion of patients with a history of diabetes in the FEVS >50% group as compared to the FEVS <40% group). In addition, a very weak positive correlation between uric acid (mg/dL) and supraventricular arrhythmia was revealed using Spearman’s correlation coefficient (r = −0.115, *p* = 0.045).

A significant statistical difference in the mean value of serum uric acid between patients in the EF >50% and EF <40% categories was observed. Patients in the EF >50% group had significantly lower uric acid levels compared to the EF <40% group of patients. A statistical difference was seen between patients in the EF 49–40% and EF <40% groups of patients, with patients in the EF 49–40% group having significantly lower uric acid levels compared to the EFS <40% group of patients.

When analyzing the NT-ProBNP levels, we noticed that patients with higher EF had lower levels of NT-proBNP. A very strong direct correlation between NT-proBNP and NYHA class was revealed using Spearman’s correlation coefficient (r = 0.811, *p* < 0.001).

Using linear regression, we obtained a significant model (*p =* 0.030), in which the variability of EF was expressed by uric acid in the proportion of approximately 1.5% (R-square = 0.015) (see Figure 1). Therefore, a significant link was revealed between FEVS and uric acid.

We also obtained significant results when a linear regression model (*p* = 0.004) was calculated, where the variability of Log NT-proBNP was expressed by uric acid in the proportion of approximately 2.7% (R-square = 0.027). Therefore, a significant link was revealed between NT-proBNP levels and serum uric acid (Figure 2).

## 6. Discussion

The proportion of patients with pefHF was higher in the analyzed group than those with mrHF and refHF, in agreement with the current change in profile of patients with heart failure. Several metanalyses showed a high heterogeneity among studies [6,7,8] when analyzing the role of uric acid in heart failure.

The focus of our study, serum uric acid, revealed statistically significant findings: patients in the pefHF group had a significantly lower level compared to the refHF and mrHF group of patients. The metabolic abnormalities that arise from the continuous reduction of ejection fraction are potential contributing causes for hyperuricemia in these patients. Uric acid, being a byproduct of purine metabolism in light of recent data, is a contributing factor to the development of heart failure, alongside other known prognostic factors in HF [9,10,11]. One interesting result is that patients with mrHF and those with pefHF showed no difference in the levels of serum uric acid, indicative of the fact that these patients have a similar metabolic profile. Huang et al. [12] showed that for every milligram of serum uric acid above the normal range, the risk for all-cause mortality increases by 4%. The single linear regression we performed showed a significant model between the variability of FEVS and uric acid, but also between NT-proBNP levels and uric acid. Data from the RELAX study [13] suggest that two-thirds of patients have a certain degree of hyperuricemia, more comorbidities, higher BNP levels, and higher C reactive protein levels. In addition, the OPT-CHF study [14] (Safety Study of Oxypurinol Added to Standard Therapy in Patients with New York Heart Association Class III–IV Congestive HF) showed that when hyperuricemia was treated, the ejection fraction increased and the clinical status improved. Though questions were raised regarding the under-optimal dose of oxypurinol in this study, beneficial effects were still observed, especially for those who did not have severely reduced ejection fraction.

When we analyzed the patients, we noticed that there were some statistical differences between the groups in terms of their baseline characteristics, which need further discussion. There were significant statistical differences in the mean values for age between patients in the FEVS >50% and FEVS <40% categories. Patients in the FEVS >50% category had a significantly higher age as compared to patients in the FEVS <40% category. Cavalcanti et al. [15] published a study in which about 400 patients were analyzed, and it was found that patients with mrHF and pefHF were older. The authors also reported that there were significantly more females than males; the same is true for this study, where there was a statistical difference between males and females, especially in the proportion of males/females in the EF >50% vs. EF <40% group and in the FEVS >50% vs. FEVS <40% group. One explanation for this difference is ischemic cardiac disease, which accounts for the majority of reduced ejection fraction patients in international registries. Other authors have found the same discrepancies when analyzing data regarding these aspects. Hao et al. [16], in a prevalence study of over 22,000 Chinese patients, found that there were more females than males (55% vs. 45%) in the pefHR group and fewer in the mrHF and refHF groups (29% and 49%, respectively).

NYHA class, although of paramount importance for patients with heart failure as a more subjective self-reporting symptom of the degree of dyspnea, is somewhat biased, with NYHA classes having a poor discriminative role in functional impairment [17]. We observed more patients reporting a class II NYHA status in the pefHF group, with a significantly lower proportion of patients with NYHA class III in the FEVS >50% category as compared to the FEVS <40% category. Overall, a statistical difference (*p* < 0.001) was observed among these patients, highlighting the heterogeneity of the parameter; explanations include exercise training, subjective reporting, underestimation of personal capabilities to perform physical tasks, and overestimation of the amount of exercise [18,19,20]. Some patients, although showing signs of congestion with edema, report a lower NYHA class. One explanation may come from their fitness status, with some being more fit than others and, implicitly, having a higher tolerance level. Another explanation is that some patients present themselves earlier when the congestion is in the early stages. Caraballo et al. [17] present results that show that the NYHA system is a weak discriminator for functional status. Because of the risk of biased results caused by the subjective nature of the NYHA system, we performed an additional analysis, where we excluded those patients with NYHA class I. We observed that there was no difference from the ANOVA one-way testing or the Bonferroni correction comparisons. The statistical testing for serum uric acid and NT-proBNP for the three groups of patients resulted in unchanged p-values from ANOVA testing.

Another somewhat important aspect was also observed when we analyzed the relationship between NYHA classes and NT-proBNP levels, with strong correlation found. A more recent study by Spinar et al. [21] revealed that, in a study based on 1088 patients with chronic heart failure, that NT-proBNP levels were a better tool for identification of high-risk patients. Univariate logistic regression showed that hyperuricemia (>8.41 mg/dL) in this studied population was one important comorbid condition, especially for the primary end-point, consisting of cardiovascular mortality (OR 3.04 (2.09; 4.43) *p* < 0.001), alongside anemia or hyponatremia.

Another study, published by Malek et al. [22] from the Acute HEart FAilure Database registry (AHEAD), which comprised patients with acute heart failure with median NT-proBNP levels of 5510 pg/ mL and a median of 8.1 mg/dL of serum uric acid, indicated that chronic treatment with allopurinol was a risk factor for long-term mortality, but was not the cause. Chronic allopurinol treatment was observed to be an identifier of patients at risk but not one of the decompensating factors. The metabolic profile of our patients was also modified. We obtained several parameters with statistical significance, such as serum potassium, creatinine levels, lipid profile, and uric acid. Patients with higher ejection fraction had significantly lower potassium in the serum compared to those who had reduced ejection fraction. One explanation is that the refHF patients had in their drug regimens potassium-sparing drugs, such as spironolactone. Savarese et al. [23] found that dyskalemia is a common finding in heart failure patients, irrespective of their ejection fraction; patients with pefHF are more susceptible to hypokalemia, and the risk of hyperkaliemia is more common in the lower ejection fraction patients. Although patients with refHF are treated with mineralocorticoid antagonists, for the rest of the patients with HF, this class of drugs is administered for other indications. There are other studies that show the benefit of mineralocorticoid antagonists in the mrHF and pefHF, with impact on reduced myocardial fibrosis, arrhythmias, brain natriuretic peptides, and the 6 min walking test [24,25]. These patients could benefit from mineralocorticoid antagonist administration, but further studies need to be conducted to correctly assess the risk associated with incidental hyperpotassemia. Our study suggests that potassium could play a role in the complete pathophysiological mechanisms underlying the development of HF.

## 7. Conclusions

These data suggest that there is a link between heart failure and the level of uric acid, where one can infer that the metabolic derangements that induce an increase in serum uric acid are likely responsible for the increase in morbidity and mortality as a result of hyperuricemia heart failure patients. In addition, in patients newly diagnosed with hyperuricemia, the risk of identifying a lower ejection fraction is higher. Single linear regression was performed to determine whether there is a direct link between these parameters, while keeping in mind that there can be several covariates that can influence this relationship. Although it is debatable whether asymptomatic hyperuricemia must be treated, underlying metabolic abnormalities are present, and there is a risk of vascular inflammation, accelerated atherosclerosis, and major cardiovascular events.

## 8. Limitations

This study has several limitations. First, this is a retrospective observational study that encompasses some 300 patients from a single center. Second, these patients had only one determination of serum uric acid during their admission to hospital, and no other value was obtained, even if patients started treatment during the admission. Third, we performed only a single linear regression between two values. Another limitation is that we included in our analysis all supraventricular arrhythmias as a single parameter, although it is highly unlikely to encounter differences between atrial fibrillation and atrial flutter when measuring serum uric acid. Patients with fewer supraventricular extrasystoles may have a different serum uric acid profile than those with atrial fibrillation, but those with higher numbers of extrasystoles could in fact resemble those with atrial fibrillation. At the same time, there was no analysis of the medication’s influence on the level of serum uric acid. All patients were not being administered serum uric acid lowering agents, and all patients received at least a dose of loop diuretic in accordance with ESC guidelines for their chronic heart failure. In our analysis, thiazide diuretics and thiazide-like diuretics, which are known to increase the levels of serum uric acid, were considered an exclusion criteria; the reason for not measuring the influence of other diuretics was to show the relationship between incidental serum uric acid and ejection fraction.

There are other parameters that influence the outcomes of patients; hyperuricemia is just one piece of the puzzle. While it may not be the most important, it is frequently unaddressed and overlooked.

## Figures and Tables

**Figure 1 jcm-10-04026-f001:**
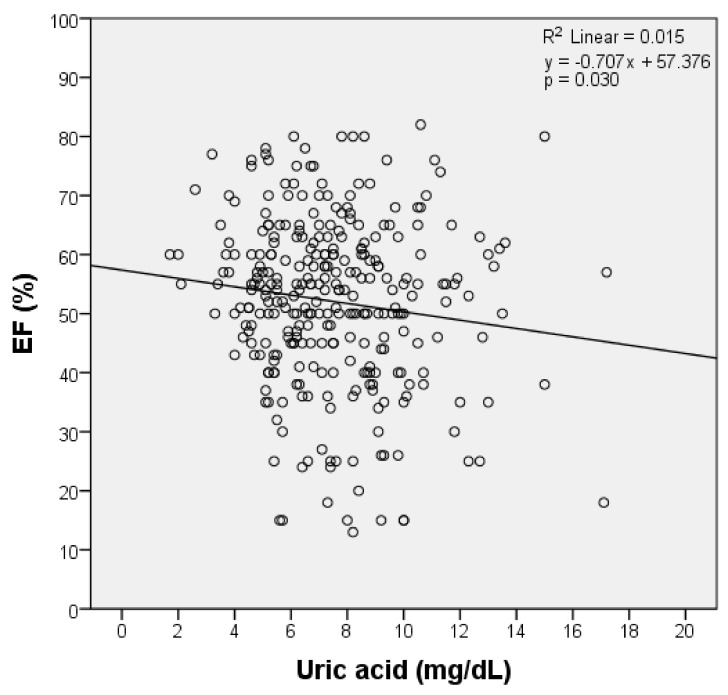
Single linear regression analysis among EF groups and serum uric acid. Scatter plot with the corresponding regression line and regression equation shows the correlation between FEVS and uric acid. Note: FEVS = left ventricular ejection fraction.

**Figure 2 jcm-10-04026-f002:**
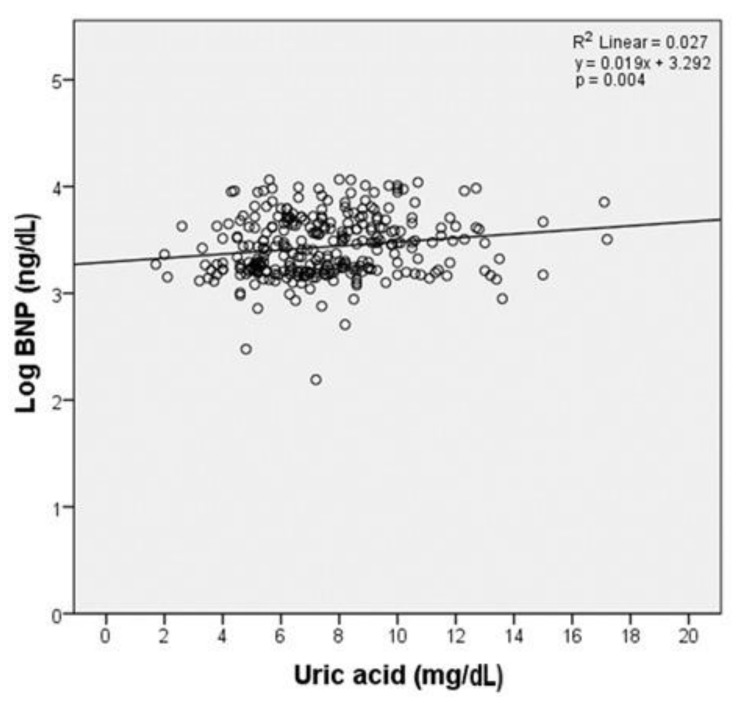
Single linear regression analysis among Log BNP and uric acid. Scatter plot with the corresponding regression line and regression equation shows the correlation between BNP and uric acid.

**Table 1 jcm-10-04026-t001:** Summary of baseline characteristics.

	Patients with EF >50% (*N* = 198)	Patients with EF 49–40% (*N* = 54)	Patients with EF <40% (*N* = 51)	*p-Value*	*p-Value 1*	*p-Value 2*	*p-Value 3*
***Age (years)***							
Mean (SD)	75.35 (9.385)	74.31 (10.731)	70.04 (12.625)	0.005	NS	S	NS
Min; Max	47; 96	46; 96	47; 94				
Median (Q1; Q3)	76.0 (69.0; 83.0)	77.0 (67.0; 82.0)	70.0 (60.5; 80.0)				
***Gender***							
Male	75 (37.88%)	27 (50.00%)	38 (74.51%)				
Female	123 (62.12%)	27 (50.00%)	13 (25.49%)	<0.001	NS	S	S
***NYHA class***							
I	2 (1.03%)	0 (0%)	0 (0%)	<0.00	NS	NoP	NoP
II	140 (70.71%)	32 (59.26%)	16 (31.37%)	1	NS	S	S
III	53 (26.77%)	19 (35.19%)	25 (49.02%)		NS	S	NS
IV	3 (1.52%)	3 (5.56%)	10 (19.61%)		NS	S	NS
***Death***							
No	186 (93.94%)	52 (96.30%)	43 (84.31%)				
Yes	12 (6.06%)	2 (3.70%)	8 (15.69%)	0.033	NS	NS	NS

**Table 2 jcm-10-04026-t002:** Summary of comorbidities.

	Patients with EF>50% (*N* = 198)	Patients with EF 49–40% (*N* = 54)	Patients with EF<40% (*N* = 51)	*p-Value*	*p-Value 1*	*p-Value 2*	*p-Value 3*
***History of diabetes***							
No	129 (65.15%)	33 (61.11%)	20 (39.22%)	0.003	NS	S	NS
Yes	69 (34.85%)	21 (38.89%)	31 (60.78%)				
***History of coronary artery disease***							
No	103 (52.02%)	25 (46.30%)	20 (39.22%)	0.243	NS	NS	NS
Yes	95 (47.98%)	29 (53.70%)	31 (60.78%)				
***History of arrhythmias***							
No	86 (43.43%)	22 (40.74%)	23 (45.10%)	0.899	NS	NS	NS
Yes	112 (56.57%)	32 (59.26%)	28 (54.90%)				

**Table 3 jcm-10-04026-t003:** Summary of main laboratory test results by FEVS (note FEVS = left ventricular ejection fraction) categories.

	Patients with EF >50% (*N* = 198)	Patients with EF 49–40% (*N* = 54)	Patients with EF <40% (*N* = 51)	*p-Value*	*p-Value 1*	*p-Value 2*	*p-Value 3*
***Na+ (mmol/L)***							
Mean (SD)	139.35 (4.950)	139.54 (5.255)	138.92 (5.176)	0.808	NS	NS	NS
Min; Max	122; 152	116; 147	124; 148				
Median (Q1; Q3)	140.0 (138.0; 142.0)	141.0 (138.0; 143.0)	140.0 (137.0; 141.5)				
***K+ (mmol/L)***							
Mean (SD)	4.327 (0.7717)	4.463 (0.7088)	4.418 (0.8021)	0.027	NS	S	NS
Min; Max	1.7; 7.5	3.3; 6.2	2.7; 6.4				
Median (Q1; Q3)	4.40 (3.90; 4.80)	4.50 (3.90; 5.00)	4.60 (3.95; 5.25)				
***Serum creatinine (mg/dL)***							
Mean (SD)	1.5941 (0.75901)	1.6957 (0.61205)	1.9080 (0.95739)	0.035	NS	S	NS
Min; Max	0.52; 4.97	0.90; 3.50	0.71; 7.23				
Median (Q1; Q3)	1.420 (1.060; 1.860)	1.585 (1.230; 1.930)	1.710 (1.445;				
***eGFR (MDRD) (mL/min***)							
Mean (SD)	46.535 (21.2707)	42.222 (16.7025)	42.143 (17.3025)	0.196	NS	NS	NS
Min; Max	8.3; 124.6	13.2; 87.2	8.3; 83.6				
Median (Q1; Q3)	43.90 (32.10; 57.70)	43.35 (28.30; 50.40)	39.90 (30.20; 51.40)				
***Serum uric acid (mg/dL)***				0.004	NS	S	S
Mean (SD)	7.328 (2.5407)	7.091 (2.0394)	7.798 (2.5859)				
Min; Max	1.7; 17.2	4.0; 12.8	5.1; 17.1				
Median (Q1; Q3)	7.05 (5.40; 8.60)	6.65 (5.40; 8.70)	8.20 (6.50; 9.90)				
***NT-proBNP***							
Mean (SD)	2422.86 (1491.644)	3699.20 (1731.444)	6667.92 (2714.724)	<0.001	S	S	S
Min; Max	155; 10209	1654; 9120	1884; 11640				
Median (Q1; Q3)	1770.0 (1524.0; 3128.0)	3075.0 (2613.0; 4800.0)	6430.0 (4617.0; 9130.0)				
***Left atrial volume (mL)***							
Mean (SD)	62.41 (10.752)	80.85 (5.839)	102.39 (12.405)	<0.001	S	S	S
Min; Max	35; 91	69; 90	84; 130				
Median (Q1; Q3)	64.0 (53.0; 70.0)	81.0 (75.0; 87.0)	99.0 (93.0; 113.5)				

## Data Availability

Data supporting reported results can be provided on request.

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
