# Peer review of "The Relationship between Serum Uric Acid and Ejection Fraction of the Left Ventricle"

_jcm, 2021, doi:10.3390/jcm10174026_

Round 1

Reviewer 1 Report

The impact of serum uric acid n heart failure is still not not known. The authors revealed the relationship between serum uric acid and the ejection fraction in 303 patients with decompensated heart failure. The study is retrospective. I have few suggestions:
I do not understand the NYHA class I and II in decompansated population - it should be explained.
One of the variable is arrhythmia - it is important to analyse atriall fibrillation vs sinus rhythm in relation to SUA.

Discussion should be rewritten: in the first line the results of SUA and EF.

I suggest the analysis of the diuretics and doses in relation to SUA level.

Reviewer 2 Report

The authors present the study about the relationship between serum uric acid and left vantricular ejection fraction (LVEF) in the group of patients with heart failure (HF). While the uric acid level in patients with HF is an important factor  of prognosis, in my opinion assessment of the relationship between uric acid serum concentration and LVEF in such heterogeneous group of patients with HF makes no sense from the clinical or even cognitive point of view. Furthermore there are major methodological limitations that may undetermine the credibility of the conclusions and the clinical implications of the work.

Specific comments in detail:

  1. The statistics are inadequate; professional statistics consultation should be considered.
  2. Professional language consultation should be considered
  3. The study group isn't precisely defined - is this group with acute or chronic HF? Furthrmore there is no data about exclusion criteria especially about states and diseases associated with higher uric acid levels such as neoplasms, connective tissue diseases or alcohol abuse etc.
  4. Baseline characteristics of the patients cohort are not clearly presented. Firstly, the study lacks the basic laboratory characteristics of patients. Secondly there is no data about the medications in analysed group of patients. What percentage of patients was treated with allopurinol? Are analysed patients treated optimally?
  5. What percentage of patients was protected against sudden cardiac death by ICD/CRT-D?
  6. The main limitation of the study is a significant disproportion in the size of the analysed group (patients with LVEF>50% n=198, patients with LVEF 49-40% n=53 and patients with LVEF<40%, n=51) which significantly affects the results.
  7. The discussion requires editing with carefu comparison of own results with other previous studies

Round 2

Reviewer 1 Report

The reviewer suggestions/remarks are included. However I strongly suggest to excluded 2 patients with NYHA I and check if the results will be changed.

Could you discuss the relationship between NYHA class and NTproBNP level in the studied population? it is more important than fit status in exacerbation of heart failure.

Reviewer 2 Report

Thank you for your answers. I have a few more comments. The discussion requires editing. The results obtained in the analysis should be removed from the "Discussion" section. The numerical values of the results should only appear in the "Results" section.
